# Characterization of Overfitting in Robust Multiclass Classification

**Jingyuan Xu**    **Weiwei Liu**[*]
School of Computer Science, Wuhan University
National Engineering Research Center for Multimedia Software, Wuhan University
Institute of Artificial Intelligence, Wuhan University
Hubei Key Laboratory of Multimedia and Network Communication Engineering, Wuhan University
{jingyuanxu777,liuweiwei863}@gmail.com

## Abstract

This paper considers the following question: Given the number of classes $m$, the number of robust accuracy queries $k$, and the number of test examples in the dataset $n$, how much can adaptive algorithms robustly overfit the test dataset? We solve this problem by equivalently giving near-matching upper and lower bounds of the robust overfitting bias in multiclass classification problems.

## 1    Introduction

Learning models that are robust to adversarial perturbations has garnered significant research attention in recent years. However, despite this progress, a pervasive issue continues to plague these models, namely *robust overfitting* [1]. A common approach to overcome overfitting is to divide the dataset into a training set, and a holdout (or test) set. Nonetheless, modern machine learning is *adaptive* in its nature. Prior information about a model's performance on the test set inevitably influences future modeling choices in extensive experiments and competitions. Recent studies have shown that excessive reuse of the holdout dataset can also leads to overfitting in non-robust setting [2, 3], and a body of subsequent work has quantitatively explored this phenomenon in the framework of *perfect test label reconstruction* [4, 5, 6]. Accordingly, in this paper we attempt to address the following questions: Can adaptive behavior result in overfitting in an adversarial setting? If so, by how much can adaptive algorithms robustly overfit the test dataset?

To solve these questions, we generalize the framework of perfect reconstruction to the adversarial setting, and analyze the average case performance that can be achieved by an adaptive algorithm, denoted as $h_{\mathcal{U}}(k, n, m)$, where $k, n, m$ represents the number of robust accuracy queries, test samples and classes, respectively. This term equivalently measures the maximum level of robust overfitting in a multiclass classification problem. In this paper, we derive both upper and lower bounds of $h_{\mathcal{U}}(k, n, m)$, and demonstrate that our upper bounds and lower bounds are matching within logarithmic factors when $n$ and the distribution of test dataset features $\mathcal{D}_{\mathcal{X}}$ are fixed.

### 1.1    Related works

**Perfect test label reconstruction.** The question of perfect test label reconstruction in non-robust setting dates back to decades ago [7, 8, 9]. The developments on studying biasing results due to adaptive reuse of the test data start with the work of [4, 10], which broadly fall in the field of adaptive data analysis [11, 12]. [5] first pose the problem of characterizing the overfitting bias as a function of $k, n, m$, but they fail to give upper and lower bounds on the same order of $m$, which is left as an open

---

[*]Correspondence to: Weiwei Liu <liuweiwei863@gmail.com>.

37th Conference on Neural Information Processing Systems (NeurIPS 2023).

question [13]. [6] close this question and determine the amount of overfitting possible in multiclass classification. The theory on quantitative overfitting analysis in adversarial setting remains blank.

**Adversarial robustness.** It has been shown that deep neural networks are fragile to imperceptible distortions in the input space [14]. Perhaps the most common method to improve adversarial robustness is adversarial training [15, 16]. Theoretically, [17, 18, 19, 20] study the PAC learnability of adversarial robust learning problem, and [21, 22] study adversarial robustness under self-supervised learning. [23, 24] investigates the adversarial robustness from the perspective of ordinary differential equations. Besides, [25] analyze the trade-off between robustness and fairness, [26] study the worst-class adversarial robustness in adversarial training.

## 2 Summary of our results

In the remainder of this article, $\mathbb{R}, \mathbb{N}$ and $\mathbb{R}^d$ represent the sets of real numbers, natural numbers and $d$-dimensional vectors over $\mathbb{R}$, respectively. We denote the set $\{1, \ldots, n\}$ (for $n \in \mathbb{N}$) by $[n]$. If $A$ and $B$ are sets, we use $B^A$ to denote the collection of all mappings from $A$ to $B$ and $2^A$ to denote the power set of $A$, that is the collection of all subsets of $A$. We denote the indicator function by $\mathbb{1}\{\text{event}\}$, that is 1 if an event happens and 0 otherwise. If $\mathcal{D}$ is a distribution, we use $\mathrm{supp}(\mathcal{D})$ to denote the support of $\mathcal{D}$, which is defined by the closure of the set of possible values of a random variable having $\mathcal{D}$ as its distribution. Besides, $\|\cdot\|_p$ represents the $\ell_p$ norm and $d_p(\cdot, \cdot)$ represents the distance function induced by $\ell_p$ norm. Finally, we use big tilde notations $\tilde{O}, \tilde{\Omega}$ and $\tilde{\Theta}$ as variants of big O notations that ignores logarithmic factors.

### 2.1 Problem formulation

Let $\mathcal{X} = \mathbb{R}^d$ be the instance space and $\mathcal{Y} = \{1, \ldots, m\}$ be the label space. Let $S = \{(x_i, c_i)\}_{i=1}^n$ denote the test set, whose features, denoted as $\mathcal{X}_S = \{x_1, \ldots, x_n\}$, are independent and identically distributed (i.i.d.) according to some distribution $\mathcal{D}_\mathcal{X}$ on $\mathcal{X}^2$. For simplification, we use $\bar{c} = (c_1, \ldots, c_n)$ to describe the vector of test set labels. Let $f : \mathcal{X} \to \mathcal{Y}$ be a function, its *robust accuracy* on the test set with respect to (w.r.t.) a small perturbation $\mathcal{U} : \mathcal{X} \to 2^\mathcal{X}$ is defined by

$$\mathrm{Acc}_\mathcal{U}(f; S) \triangleq \frac{1}{n} \sum_{i=1}^n \mathbb{1}\{\forall x' \in \mathcal{U}(x_i), f(x') = c_i\}.$$

The perturbation $\mathcal{U}(x)$ is required to be nonempty, so some choice of $x'$ is always available. This paper focuses the case when the perturbation is the $p$-norm ball with a small radius $r$, i.e. $\mathcal{U}(x) = \{z \in \mathcal{X} : \|z - x\|_p \le r\}$ for some $p \ge 1$. $r = 0$ gives the *identity perturbation*: $\mathcal{I}(x) \triangleq \{x\}$. Note that in this case, the definition of robust accuracy is reduced to standard (non-robust) accuracy.

In this work, we mainly study the robust overfitting attack algorithms, which do not have access to the test set $S$. Instead, they have query access to robust accuracy of the model on $S$, that is, for any classifier $f$, the algorithm is able to obtain the value $\mathrm{Acc}_\mathcal{U}(f; S)$. We refer to this access as a *query*. A *k-query algorithm* $\mathcal{A}$ makes $k$ queries $f_1, \ldots, f_k$ on $S$, and based on the values of $\mathcal{X}_S$ and $\mathrm{Acc}_\mathcal{U}(f_1; S), \ldots, \mathrm{Acc}_\mathcal{U}(f_k; S)$, $\mathcal{A}$ outputs a classifier $\hat{f} = \mathcal{A}(S)$. We say a $k$-query algorithm $\mathcal{A}$ is based on hypothesis class $\mathcal{H} \subset \mathcal{Y}^\mathcal{X}$ if $f_1, \ldots, f_k \in \mathcal{H}$. The performance of the algorithm on $S$ is measured by

$$h_\mathcal{U}(\mathcal{A}; S) \triangleq \mathbb{E}\left[\mathrm{Acc}_\mathcal{U}(\hat{f}; S)\right],$$

where the expectation is over the algorithm's randomization. It is also of interest to ask the performance of an algorithm when $\bar{c}$ are drawn according to some distribution $\mathcal{D}_\mathcal{Y}$ over $[m]^n$. Let $\mathcal{D} = \mathcal{D}_\mathcal{X} \times \mathcal{D}_Y$, for an algorithm $\mathcal{A}$, define its performance w.r.t. $\mathcal{D}$ by

$$h_\mathcal{U}(\mathcal{A}; \mathcal{D}) \triangleq \mathbb{E}_{S \sim \mathcal{D}} h_\mathcal{U}(\mathcal{A}; S).$$

And we evaluate the algorithms under the assumption that they do not have any prior knowledge about the test labels. That is, the prior distribution of test labels is uniform over all possible labeling:

$$h_\mathcal{U}(\mathcal{A}) \triangleq \mathbb{E}_{S \sim \mathcal{X}_S^n \times \mu_m^n} h_\mathcal{U}(\mathcal{A}; S),$$

---

[2]Formally, there is a sigma algebra $\mathcal{F} \subset 2^\mathcal{X}$ of events and $\mathcal{D}_\mathcal{X}$ is a probability measure on $(\mathcal{X}, \mathcal{F})$

where $\mu_m^n$ denotes the uniform distribution over $[m]^n$. Since the best robust accuracy is $1/m$ without making any queries ($k = 0$), $h_{\mathcal{U}}(\mathcal{A}; S) - 1/m$ measures how much $\mathcal{A}$ robustly overfits $S$. The goal of this paper is to find the largest robust overfitting possible for a multiclass classification problem. So we define the performance that is achievable by an algorithm after making $k$ queries on any $S$ by

$$h_{\mathcal{U}}(k, n, m) \triangleq \max_{\mathcal{A}} h_{\mathcal{U}}(\mathcal{A}),$$

and we are interested in the value of $h_{\mathcal{U}}(k, n, m) - 1/m$, or $h_{\mathcal{U}}(k, n, m)$ equivalently. In the rest of this paper, we aim at deriving bounds of $h_{\mathcal{U}}(k, n, m)$ for given $k, n$ and $m$.

## 2.2 Our main results

Our bounds on $h_{\mathcal{U}}(k, n, m)$ have two different regimes, which can be summarized as the following theorem.

**Theorem 1** (Informal). *let $\mathcal{D}_{\mathcal{X}}$ be the distribution of test sample features. For $k = \tilde{O}(n/m)$,*

$$\Phi_{\mathcal{D}_{\mathcal{X}}}(n) \cdot \left[ \frac{1}{m} + \Omega\left( \sqrt{\frac{k}{mn}} \right) \right] \leq h_{\mathcal{U}}(k, n, m) \leq \frac{1}{m} + \tilde{O}\left( \sqrt{\frac{k}{nm}} \right).$$

*For $k = \tilde{\Omega}(n/m)$,*

$$\Phi_{\mathcal{D}_{\mathcal{X}}}(n) \cdot \left[ \frac{1}{m} + \tilde{\Omega}\left( \frac{k}{n} \right) \right] \leq h_{\mathcal{U}}(k, n, m) \leq \frac{1}{m} + \tilde{O}\left( \frac{k}{n} \right),$$

*where $\Phi_{\mathcal{D}_{\mathcal{X}}}(n) \leq 1$ and is monotonically decreasing w.r.t. $n$.*

When $\mathcal{U} = \mathcal{I}$, our upper bounds match the best known upper bounds [5], while our lower bounds differ from the known optimal lower bounds [6] by a factor $\Phi_{\mathcal{D}_{\mathcal{X}}}(n)$. Since overfitting in the context of robust learning has some requirements on the test samples (e.g. the well-separated property for different classes [27]), we may not able to ensure a non-trivial lower bound on $h_{\mathcal{U}}(k, m, n)$ for some $\mathcal{D}_{\mathcal{X}}$. Intuitively, $\Phi_{\mathcal{D}_{\mathcal{X}}}(n)$ measures how easily to sample $n$ 'good' (for robust overfitting) test data features from $\mathcal{D}_{\mathcal{X}}$. The specific form of $\Phi_{\mathcal{D}_{\mathcal{X}}}(n)$ is presented in Section 3.2. Note that for a fixed size of $S$ whose features are i.i.d. according to a fixed distribution $\mathcal{D}_{\mathcal{X}}$, the upper and lower bounds are matching up to a logarithmic factor, that is,

$$h_{\mathcal{U}}(k, n, m) = \tilde{\Theta}\left( \frac{1}{m} + \sqrt{\frac{k}{mn}} \right), \qquad k = \tilde{O}(n/m),$$

and

$$h_{\mathcal{U}}(k, n, m) = \tilde{\Theta}\left( \frac{1}{m} + \frac{k}{n} \right), \qquad k = \tilde{\Omega}(n/m)$$

for any fixed $n$ and $\mathcal{D}_{\mathcal{X}}$.

## 2.3 Overview of our techniques

Next, we give a brief overview of proof techniques used to obtain the main results. We first note that throughout this paper we use the notion of corrupted hypothesis [28], which transforms the formulation of robust accuracy to a non-robust one thus greatly simplifying the proofs. The definition of corrupted hypothesis is presented in the beginning of Section 3.

- We establish the upper bounds via minimum description length argument, following closely a proof of an analogous result by [5] for non-robust setting. Note that their bounds can be viewed as trivial upper bounds of $h_{\mathcal{U}}(k, n, m)$ since non-robust accuracy always upper bounds robust accuracy. We tighten the bounds by considering the query class of an algorithm. The details are presented in Theorem 2.

- To obtain the lower bounds, we propose computationally efficient algorithms for two regions of $k$ respectively. The algorithms are modified from [6], who study the worst case overfitting bias in a non-robust setting. To take the perturbation of features into account, we extend their queries to the whole feature space $\mathcal{X}$ by assigning each label of $x \in \mathcal{X}$ to be the 'closest' label in $\mathcal{X}_S$ in the sense of $p$-norm. The theoretical guarantees are given in Theorems 3 and 4.

## 2.4 Discussion

Although we equivalently derive both upper bounds and lower bounds of robust overfitting bias, there remains a gap in finding whether the lower and upper bounds can match up to a constant factor. In non-robust setting, it is proven that the $\sqrt{\log n}$ factor in upper bounds can be removed for $k = 1$ [6]. It is interesting to ask if this result holds for adversarial robust setting and, more ambitiously, for all $k$. We leave it as an open question.

## 3 Proofs of main results

To make the proofs more readable, we introduce the notion of corrupted hypothesis [28].

Consider a given hypothesis $f : \mathcal{X} \to \mathcal{Y}$. A labeled adversarial sample $(\tilde{x}, y)$ is classified correctly if $\tilde{x} \in f^{-1}(y)$. A labeled example $(x, y)$ is classified correctly if $\mathcal{U}(x) \subset f^{-1}(y)$. Let $\tilde{\mathcal{Y}} = \mathcal{Y} \cup \{\perp\}$, where $\perp$ is the special "always wrong" output, i.e. $y \neq \perp$ for every $y \in \tilde{\mathcal{Y}}$. We define the mapping $\kappa_{\mathcal{U}} : \mathcal{Y}^{\mathcal{X}} \to \tilde{\mathcal{Y}}^{\mathcal{X}}$:

$$\kappa_{\mathcal{U}}(f)(x) = \begin{cases} y, & \mathcal{U}(x) \subset f^{-1}(y), \\ \perp, & \text{otherwise.} \end{cases}$$

The corrupted set of hypotheses induced by perturbation $\mathcal{U}$ is then defined by $\tilde{\mathcal{H}} = \{\kappa_{\mathcal{U}}(f) : f \in \mathcal{H}\}$.

### 3.1 Upper bounds

The upper bounds relies on the following theorem, showing in a probabilistic view that finding a classifier in some given hypothesis class with desired robust accuracy requires learning many bits about the test labeling.

**Theorem 2.** *Let $m, n, k$ be positive integers and $\mathcal{D}_m^n = \mathcal{D}_{\mathcal{X}}^n \times \mu_m^n$, where $\mu_m^n$ denotes the uniform distribution over $[m]^n$. Let $\mathcal{H} \subset \mathcal{Y}^{\mathcal{X}}$ be a hypothesis class and $\mathcal{U}$ be an perturbation. Then there exists a constant $C_{\mathcal{H}} \leq 1$ satisfying: For every $\mathcal{H}$-based $k$-query algorithm $\mathcal{A}, \delta > 0, b = k \ln(n + 1)1 + \ln(1/\delta)$, we have*

$$\Pr_{S \sim \mathcal{D}_m^n, f = \mathcal{A}(S)} \left\{ \mathrm{Acc}_{\mathcal{U}}(f; S) \geq \frac{C_{\mathcal{H}}}{m} + 2\sqrt{\frac{b}{nm}} \right\} \leq \delta, \qquad k < \frac{n}{m(\log(n + 1) + \log(1/\delta))},$$

*and*

$$\Pr_{S \sim \mathcal{D}_m^n, f = \mathcal{A}(S)} \left\{ \mathrm{Acc}_{\mathcal{U}}(f; S) \geq \frac{C_{\mathcal{H}}}{m} + \frac{2b}{n} \right\} \leq \delta, \qquad k \geq \frac{n}{m(\log(n + 1) + \log(1/\delta))}.$$

*Proof.* For any fixed hypothesis $f$, denote $\kappa_{\mathcal{U}}(f)$ by $\tilde{f}$. By the definition of corrupted hypotheses, for every $i \in [n]$,

$$\Pr_{x_i \sim \mathcal{D}_{\mathcal{X}}} \{\forall x' \in \mathcal{U}(x_i), f(x') = c_i\} = \Pr_{x_i \sim \mathcal{D}_{\mathcal{X}}} \{\tilde{f}(x) = c_i\}$$

$$= \Pr_{x_i \sim \mathcal{D}_{\mathcal{X}}} \{\tilde{f}(x_i) = c_i | \tilde{f}(x_i) \neq \perp\} \Pr_{x_i \sim \mathcal{D}_{\mathcal{X}}} \{\tilde{f}(x_i) \neq \perp\}$$

$$+ \Pr_{x_i \sim \mathcal{D}_{\mathcal{X}}} \{\tilde{f}(x_i) = c_i | \tilde{f}(x_i) = \perp\} \Pr_{x_i \sim \mathcal{D}_{\mathcal{X}}} \{\tilde{f}(x_i) = \perp\}.$$

We observe that $\Pr\{\tilde{f}(x_i) \neq \perp\}$ is a constant related to $f$, let $C(f) \triangleq \Pr\{\tilde{f}(x_i) \neq \perp\}$, since $\Pr\{\tilde{f}(x_i) = c_i | \tilde{f}(x_i) = \perp\} = 0$, we have

$$\Pr_{x_i \sim \mathcal{D}_{\mathcal{X}}} \{\forall x' \in \mathcal{U}(x_i), f(x') = c_i\} = \Pr_{x_i \sim \mathcal{D}_{\mathcal{X}}} \{\tilde{f}(x_i) = c_i | \tilde{f}(x_i) \neq \perp\} \Pr_{x_i \sim \mathcal{D}_{\mathcal{X}}} \{\tilde{f}(x_i) \neq \perp\} = \frac{C(f)}{m}.$$

This implies that $\mathbb{1}\{\forall x' \in \mathcal{U}(x_i), f(x') = c_i\}$ is a Bernoulli random variables with bias $\frac{C(f)}{m}$. By the Chernoff bound, for any fixed $f$,

$$\Pr_{S \sim \mathcal{D}_m^n} \left\{ \frac{1}{n} \sum_{i=1}^{n} \mathbb{1}\{\forall x' \in \mathcal{U}(x_i), f(x') = c_i\} \geq \frac{C(f)}{m} + \epsilon \right\} \leq e^{-\frac{mn\epsilon^2}{2C(f) + m\epsilon}}.$$

Denote $\max_{f\in\mathcal{H}} C(f)$ by $C_\mathcal{H}$, then

$$\mathbf{Pr}\left\{\mathrm{Acc}_\mathcal{U}\left(f;S\right)\geq\frac{C_\mathcal{H}}{m}+\epsilon\right\}\leq\mathbf{Pr}\left\{\mathrm{Acc}_\mathcal{U}\left(f;S\right)\geq\frac{C(f)}{m}+\epsilon\right\}\leq e^{-\frac{mn\epsilon^2}{2C(f)+m\epsilon}}\leq e^{-\frac{mn\epsilon^2}{2C_\mathcal{H}+m\epsilon}},\quad(1)$$

holds for every $f$. Consider the execution of $\mathcal{A}$ with responses of robust accuracy fixed to some sequence of values $\alpha=(\alpha_1,\ldots,\alpha_k)\in\{0,1/n,\ldots,1\}^k$. Denote the resulting predictor by $\mathcal{A}^\alpha$, its output distribution is fixed, hence by Eq.(1), we have

$$\mathbf{Pr}_{S\sim\mathcal{D}_m^n,f=\mathcal{A}^\alpha}\left\{\mathrm{Acc}_\mathcal{U}\left(f;S\right)\geq\frac{C_\mathcal{H}}{m}+\epsilon\right\}\leq e^{-\frac{mn\epsilon^2}{2C_\mathcal{H}+m\epsilon}}.$$

Denote the set of all possible values of $\alpha$ by $V$. For every test set $S$, the accuracy oracle outputs some responses in $V$. Therefore,

$$\mathbf{Pr}_{S\sim\mathcal{D}_m^n,f=\mathcal{A}(S)}\left\{\mathrm{Acc}_\mathcal{U}\left(f;S\right)\geq\frac{C_\mathcal{H}}{m}+\epsilon\right\}$$

$$\leq\sum_{\alpha\in V}\mathbf{Pr}_{S\sim\mathcal{D}_m^n,f=\mathcal{A}^\alpha}\left\{\mathrm{Acc}_\mathcal{U}\left(f;S\right)\geq\frac{C_\mathcal{H}}{m}+\epsilon\right\}$$

$$\leq(n+1)^k\cdot e^{-\frac{mn\epsilon^2}{2C_\mathcal{H}+m\epsilon}}.$$

Now if $\frac{k\ln(n+1)+\ln(1/\delta)}{n}\geq\frac{1}{m}$, then by definition of $b$, $\frac{2b}{n}\geq\frac{2}{m}$. It follows that $m\epsilon\geq2\geq2C_\mathcal{H}$, hence $\frac{mn\epsilon^2}{2C_\mathcal{H}+m\epsilon}\geq\frac{n\epsilon}{2}$ and

$$(n+1)^k\cdot e^{-\frac{mn\epsilon^2}{2C_\mathcal{H}+m\epsilon}}\leq e^{k\ln(n+1)-\frac{n\epsilon}{2}}=e^{\ln\delta}=\delta.$$

If $\frac{k\ln(n+1)+\ln(1/\delta)}{n}<\frac{1}{m}$, in this case $2\sqrt{\frac{b}{nm}}\leq\frac{2}{m}$. We obtain that $m\epsilon<2$, thus $\frac{mn\epsilon^2}{2C_\mathcal{H}+m\epsilon}\geq\frac{mn\epsilon^2}{4}$ and

$$(n+1)^k\cdot e^{-\frac{mn\epsilon^2}{2C_\mathcal{H}+m\epsilon}}\leq e^{k\ln(n+1)-\frac{mn\epsilon^2}{4}}=e^{\ln\delta}=\delta,$$

and we complete the proof. $\square$

The corollary below follows immediately from Theorem 2, which gives the upper bounds of $h_\mathcal{U}(k,n,m)$.

**Corollary 1.** *Let $m,n,k$ be positive integers and $\mathcal{U}$ be a perturbation, then*

$$h_\mathcal{U}(k,n,m)\leq\frac{1}{m}+\frac{1}{n}+2\sqrt{\frac{(k+1)\log(n+1)}{nm}},\qquad k<\frac{n}{m(\log(n+1)+\log(n))}$$

*and*

$$h_\mathcal{U}(k,n,m)\leq\frac{1}{m}+\frac{2(k+1)\log(n+1)+1}{n},\qquad k\geq\frac{n}{m(\log(n+1)+\log(n))}$$

*Proof.* Denote $\mathrm{Acc}_\mathcal{U}\left(f;S\right)\triangleq X\in[0,1]$. Substitute $\delta=1/n$ in Theorem 2 and notice that $C_\mathcal{H}\leq1$ for any hypothesis class $\mathcal{H}$, hence for $k<\frac{n}{m(\log(n+1)+\log(n))}$ we have

$$\mathbf{Pr}\left\{X\geq\frac{1}{m}+2\sqrt{\frac{(k+1)\log(n+1)}{nm}}\right\}\leq\frac{1}{n}.\qquad(2)$$

Let $c=\frac{1}{m}+2\sqrt{\frac{(k+1)\log(n+1)}{nm}}$, it remains to show $\mathbb{E}X\leq c+1/n$. It is trivial for $c\geq1$. For the case that $c<1$, let $P_X$ be the probability distribution of $X$, by the definition of expectation,

$$\mathbb{E}X=\int_0^1 XdP_X=\int_0^c XdP_X+\int_c^1 XdP_X\leq c\int_0^c dP_X+\int_c^1 dP_X\leq c+\frac{1}{n},$$

where we use the fact that $\int_c^1 dP_X\leq1/n$ by Eq.(2) in the last step. The case of $k\geq\frac{n}{m(\log(n+1)+\log(n))}$ can be proved using similar arguments, and we complete the proof. $\square$

## 3.2 Lower bounds

The lower bounds of $h_{\mathcal{U}}(k, n, m)$ are derived from two designed algorithms, namely $\mathcal{A}^{small}$ and $\mathcal{A}^{big}(C)$. $\mathcal{A}^{small}$ is divided into two cases based on whether $k = 1$, and the precise analysis is presented in Theorem 3. $\mathcal{A}^{big}(C)$ accepts a parameter $C$ for calculating an intermediate variable, and we gain our Theorem 4 by setting $C = \Phi_{\mathcal{D}_{\mathcal{X}}}(n)$. Finally, the lower bounds of $h_{\mathcal{U}}(k, n, m)$ follow from

$$h_{\mathcal{U}}(k, n, m) = \max_{\mathcal{A}} h_{\mathcal{U}}(\mathcal{A}) \geq h_{\mathcal{U}}(\mathcal{A}^{small} \text{ or } \mathcal{A}^{big}(\Phi_{\mathcal{D}_{\mathcal{X}}}(n))).$$

To simplify the expression, we first introduce some definitions. For each $\mathcal{X}_S \in \mathcal{X}^n$, define

$$\Gamma_S = \left\{ x \in \mathcal{X} \middle| \exists j_1 \neq j_2 \in [n] \text{ s.t. } d_p(x, x_{j_1}) = d_p(x, x_{j_2}) = \min_{i \in [n]} d_p(x, x_i) \right\},$$

and

$$\Phi_{\mathcal{D}_{\mathcal{X}}}(n) = \min_{\mathcal{X}_S \in \text{supp}(\mathcal{D}_{\mathcal{X}}^n)} \int_{\mathcal{U}(\Gamma_S)} dP_{\mathcal{D}_{\mathcal{X}}},$$

where $\mathcal{U}(\Gamma_S) = \{\mathcal{U}(x) | x \in \Gamma_S\}$ and $P_{\mathcal{D}_{\mathcal{X}}}$ is the distribution function of $\mathcal{D}_{\mathcal{X}}$.

With these definitions, we present that:

---

**Algorithm 1** $\mathcal{A}^{small}$ ($k = 1$)

    1. Let $f^1(x)$ be the all one query, i.e. $f^1(x) = 1, \forall x \in \mathcal{X}$.

    2. Output $\hat{f}$ that

$$\hat{f}(x) = \begin{cases} 1 & \forall x \in \mathcal{X}, \quad \text{if } \text{Acc}_{\mathcal{U}}(f^1, S) \geq 1/m, \\ 2 & \forall x \in \mathcal{X}, \quad \text{otherwise.} \end{cases}$$

---

**Theorem 3.** *Let $n \geq m$ and $\mathcal{D}_m^n = \mathcal{D}_{\mathcal{X}}^n \times \mu_m^n$. Then for $1 \leq k \leq 1 + n/2m$,*

$$h_{\mathcal{U}}(\mathcal{A}^{small}, \mathcal{D}_m^n) \geq \frac{\Phi_{\mathcal{D}_{\mathcal{X}}}(n)}{m} + \frac{1}{8}\sqrt{\frac{\Phi_{\mathcal{D}_{\mathcal{X}}}(n) \cdot k}{mn}}.$$

*Proof.* Case: $k = 1$ : For $l \in \{1, \ldots, m\}$, let $N_l$ be the number of examples with label $l$. Since the labels are uniformly distributed and $\hat{f}$ is always a constant predictor, $(N_1, \ldots, N_m)$ follows a multinomial distribution with parameters $(n; \frac{1}{m}, \ldots, \frac{1}{m})$. Then by the construction of $\hat{f}$, the number of robustly and correctly predicted labels is then given by $N_1 \cdot \mathbb{1}\{N_1 \geq n/m\} + N_2 \cdot \mathbb{1}\{N_1 < n/m\}$, and the expected robust accuracy is given by

$$\mathop{\mathbb{E}}_{\bar{c} \sim \mu_m^n} [h_{\mathcal{U}}(\mathcal{A}^{small}; S)] = \frac{1}{n} \mathbb{E}[N_1 \cdot \mathbb{1}\{N_1 \geq n/m\} + N_2 \cdot \mathbb{1}\{N_1 < n/m\}].$$

We conclude that

$$\mathop{\mathbb{E}}_{\bar{c} \sim \mu_m^n} [h_{\mathcal{U}}(\mathcal{A}^{small}; S)] \geq \frac{1}{m} + \frac{1}{4}\sqrt{\frac{1}{mn}}$$

by citing the following lemma:

**Lemma 1** (Lemma 4 in [6].). *Let $n \geq m \geq 2$. If $(N_1, \ldots, N_m)$ is distributed according to a multinomial distribution with parameters $(n; \frac{1}{m}, \ldots, \frac{1}{m})$, then*

$$\mathbb{E}[N_1 \cdot \mathbb{1}\{N_1 \geq n/m\} + N_2 \cdot \mathbb{1}\{N_1 < n/m\}] \geq \frac{n}{m} + \frac{1}{4}\sqrt{\frac{n}{m}}.$$

Case: $k > 1$ : For a fixed $S$, and hence, a fixed $\hat{f} = \mathcal{A}^{small}(S)$, for each $l \in \{1, \ldots, m, \perp\}$, let $N_{i,l}^S$ be the number of examples in $B_i$ s.t. $\kappa_{\mathcal{U}}(\hat{f})(x) = l$. Since whether $\kappa_{\mathcal{U}}(\hat{f})(x)$ equals to '$\perp$' is only depends on the distribution of $x$, we have that $(N_{i,1}^S, \ldots, N_{i,m}^S, N_{i,\perp}^S)$ follows a multinomial distribution with parameters $(|B_i|; \frac{C(S)}{m}, \ldots, \frac{C(S)}{m}, 1 - C(S))$, where $C(S) = \mathbf{Pr}\{\kappa_{\mathcal{U}}(\hat{f})(x) \neq \perp\}$.

**Algorithm 2** $\mathcal{A}^{small}$ $(k > 1)$

1. Divide $[n]$ into $k-1$ blocks $B_1, \ldots, B_{k-1}$ such that $\frac{n}{k-1} \le |B_i| \le \frac{n}{2(k-1)}$ for each $i$.

2. For $1 \le i \le k$, define query $f_i$ satisfying:

$$f_i(x_j) = \begin{cases} 1, & \text{if } j \in B_1 \cup B_2 \cup \cdots \cup B_{i-1} \\ 2, & \text{otherwise.} \end{cases}$$

for $x_j \in \mathcal{X}_S$ and

$$f_i(x) = f_i(x_{j'}), \quad \text{where } j' = \arg\min_{j \in [n]} d_p(x, x_j)$$

for $x \notin \mathcal{X}_S$.

3. Output $\hat{f}$, it predicts $x_j$ by the following:

$$\hat{f}(x_j) = \begin{cases} 1 \text{ for all } j \in B_i, & \text{if } \mathrm{Acc}_{\mathcal{U}}\left(f_{i+1}, S\right) \ge \mathrm{Acc}_{\mathcal{U}}\left(f_i; S\right), \\ 2 \text{ for all } j \in B_i, & \text{otherwise.} \end{cases}$$

and predicts the rest of $x$ by $\hat{f}(x) = \hat{f}(x_{j'})$, where $j' = \arg\min_{i \in [n]} d_p(x, x_j)$.

---

Then our predictions robustly and correctly predicts $\max\{N_{i,1}^S, N_{i,2}^S\}$ examples in $B_i$ and hence $h_{\mathcal{U}}(\mathcal{A}; S) = \frac{1}{n} \sum_{i=1}^{k-1} \max\{N_{i,1}^S, N_{i,2}^S\}$. We will show it later that

$$\mathbb{E}\left[\max\{N_{i,1}^S, N_{i,2}^S\}\right] \ge \frac{C(S)|B_i|}{m} + \frac{1}{4}\sqrt{\frac{C(S)|B_i|}{m}}.$$

By summing over the blocks, we can lower bound the expected total number of robustly correct predictions made by $\mathcal{A}^{small}$ :

$$\sum_{i=1}^{k-1} \mathbb{E}\left[\max\{N_{i,1}^S, N_{i,2}^S\}\right] \ge \frac{C(S) \cdot n}{m} + \frac{k-1}{4}\sqrt{\frac{C(S) \cdot n}{2(k-1)m}} \ge \frac{C(S) \cdot n}{m} + \frac{1}{8}\sqrt{\frac{C(S) \cdot nk}{m}}.$$

To obtain a lower bound of $C(S)$ that is independent of the choice of $S$, for each $S$ we define $g_S$ satisfying:

$$g_S(x) = \begin{cases} c_i & \forall x \in \mathcal{X}_S, \\ g_S(x_{j'}) & \text{where } j' = \arg\min_{i \in [n]} d_p(x, x_j) \quad \text{otherwise,} \end{cases}$$

and let $\mathcal{H}_n = \{g_S : S \in (\mathrm{supp}(\mathcal{D}_{\mathcal{X}}) \times \mathcal{Y})^n\}$. It is easy to see that $\mathcal{A}^{small}(S) \in \mathcal{H}_n$ for all $S$. Hence by the definition of $\Phi_{\mathcal{D}_{\mathcal{X}}}(n)$, we have

$$\Phi_{\mathcal{D}_{\mathcal{X}}}(n) = \min_{g_S \in \mathcal{H}_n} \mathbf{Pr}\left\{\kappa_{\mathcal{U}}(g_s)(x) \ne \perp\right\} \le C(S).$$

We conclude that

$$\sum_{i=1}^{k-1} \mathbb{E}\left[\max\{N_{i,1}^S, N_{i,2}^S\}\right] \ge \frac{\Phi_{\mathcal{D}_{\mathcal{X}}}(n) \cdot n}{m} + \frac{1}{8}\sqrt{\frac{\Phi_{\mathcal{D}_{\mathcal{X}}}(n) \cdot nk}{m}}.$$

Normalizing by $n$ proves the desired result.
It remains to lower bound $\mathbb{E}\left[\max\{N_{i,1}^S, N_{i,2}^S\}\right]$. Since $N_{i,1}^S$ and $N_{i,2}^S$ follow a binominal distribution with parameters $(|B_i|; \frac{C(S)}{m})$, $E[N_{i,1}^S + N_{i,2}^S] = \frac{2C(S)|B_i|}{m}$ for all $S$. Let $N'$ be an independent copy

of $N_{i,2}^S$. $N_{i,1}^S$ and $N'$ are negatively correlated, hence

$$\mathbb{E}\left[|N_{i,1}^S - N_{i,2}^S|\right] \geq \mathbb{E}\left[|N_{i,1}^S - N'|\right] \geq \mathbb{E}\left[N_{i,1}^S - \frac{C(S)}{m}|B_i|\right]$$

$$\geq \sqrt{\frac{\mathbb{E}\left[(N_{i,1}^S - \frac{C(S)}{m}|B_i|)^2\right]}{2}}$$

$$\geq \sqrt{\frac{C(S)|B_i|}{2m}(1 - \frac{C(S)}{m})}$$

$$\geq \sqrt{\frac{C(S)|B_i|}{4m}}$$

For all $S$.

Then we conclude that

$$\max\{N_{i,1}^S, N_{i,2}^S\} = \frac{N_{i,1}^S + N_{i,2}^S}{2} + \frac{|N_{i,1}^S - N_{i,2}^S|}{2} \geq \frac{C(S)|B_i|}{m} + \frac{1}{4}\sqrt{\frac{C(S)|B_i|}{m}},$$

which completes the proof. $\qquad\square$

We then present $\mathcal{A}^{big}(C)$ and the theoretical analysis for $k = \Omega(n/m)$.

---

**Algorithm 3** $\mathcal{A}^{big}(C)$

---

1. Let $t := 1 + \frac{Ck}{9\ln m}$
2. Define query $f_1, \ldots, f_k$ satisfying:
   (a) For $1 \leq j \leq t$, $f_1(x_j), \ldots, f_k(x_j)$ are uniformly chosen from all sequences in $[m]^k$ that have each element in $[m]$ appearing exactly $k/m$ times.
   (b) For $j > t$, $f_i(x_j) = \perp$ for all $i = 1, \ldots, k$.
   (c) For $x \notin \mathcal{X}_S$, $f_i(x) = f_i(x_{j'})$ for each $i \in [k]$, where $j' = \arg\min_{i \in [n]} d_p(x, x_j)$.
3. Output $\hat{f}$ such that:

$$\hat{f}(x_j) = \begin{cases} \arg\max_{y \in [m]} \sum_{i:f_i(x_j)=y} \text{Acc}_{\mathcal{U}}(f_i; S)^3, & 1 \leq j \leq t, \\ 1, & j > t. \end{cases}$$

and predicts the rest of $x$ by $\hat{f}(x) = \hat{f}(x_{j'})$, where $j' = \arg\min_{i \in [n]} d_p(x, x_j)$.

---

**Theorem 4.** *Let* $k > \frac{18m\log m}{\Phi_{\mathcal{D}_{\mathcal{X}}}(n)}$ *and let* $\mathcal{D}_m^n = \mathcal{D}_{\mathcal{X}}^n \times \mu_m^n$,

$$h_{\mathcal{U}}(\mathcal{A}^{big}(\Phi_{\mathcal{D}_{\mathcal{X}}}(n)), \mathcal{D}_m^n) \geq \frac{\Phi_{\mathcal{D}_{\mathcal{X}}}(n)}{m} + \frac{\Phi_{\mathcal{D}_{\mathcal{X}}}(n) \cdot k}{144n\log m}.$$

*Proof.* For $l \in [m]$, let $A_l$ be the total number of robustly and correctly predicted examples by all the queries that predict the first examples as '$l$', i.e.

$$A_l := n \cdot \sum_{i:\forall x' \in \mathcal{U}(x_1), f_i(x')=l} \text{Acc}_{\mathcal{U}}(f_i; S) = \sum_{i:\forall x' \in \mathcal{U}(x_1), f_i(x')=l} \sum_{j=1}^n \mathbb{1}\{f_i(x') = y_j, \forall x' \in \mathcal{U}(x_j)\}$$

$$= W_0 \cdot \mathbb{1}\{l = y_1\} + \sum_{i:\forall x' \in \mathcal{U}(x_1), f_i(x')=l} \sum_{j=2}^t \mathbb{1}\{f_i(x') = y_j, \forall x' \in \mathcal{U}(x_j)\},$$

(3)

where $W_0 \leq \frac{k}{m}$ is the number of queries that predict the whole perturbation set $\mathcal{U}(x_1)$ as '$y_1$'.

---

[2] Breaking ties randomly.

Let
$$M_l := \sum_{i:\forall x' \in \mathcal{U}(x_1), f_i(x')=l} \sum_{j=2}^{t} \mathbb{1}\{f_i(x') = y_i, \forall x' \in \mathcal{U}(x_i)\},$$

then for $l \neq y_1$,
$$A_{y_1} - A_l = W_0 + M_{y_1} - M_l.$$

Since $\mathbb{1}\{f_i(x') = y_j, \forall x' \in \mathcal{U}(x_j)\}$ are independent for $j \neq j'$, and are negatively associated across $i$ for any fixed $j$, therefore, by the Chernoff bound, it satisfies that for $\epsilon < 1$

$$\mathbf{Pr}\{|M_l - \mathbb{E}[M_l]| > \epsilon\,\mathbb{E}[M_l]\} \leq 2\exp(-\frac{\epsilon^2}{3}\mathbb{E}[M_l]).$$

Suppose $\epsilon$ satisfies that $\epsilon\,\mathbb{E}[M_l] \leq \frac{k}{2m}$ and $\frac{\epsilon^2}{3}\mathbb{E}[M_l] \geq 3\log m$, then by Eq. (3) and the union bound

$$\mathbf{Pr}\left\{\arg\max_{l \in [m]} A_l \neq y_1\right\} < (m-1) \cdot \frac{2}{m^3} \leq \frac{1}{4},$$

and with probability at least $3/4$, $\hat{f}(x_1) = y_1$.

We now derive the bounds on $\epsilon$. To this end, we first need to bound $\mathbb{E}[M_l]$. Define $\mathcal{H}_n$ as that in the proof of Theorem 3. Let $\widetilde{\mathcal{H}_n}$ denotes the corrupted set of $\mathcal{H}_n$. For each $\tilde{g} \in \widetilde{\mathcal{H}_t}$, define $\tilde{g}_\perp : \mathcal{X} \to \tilde{\mathcal{Y}}$ that takes value '$\perp$' on $\mathcal{U}(x_j)$ for $j = t+1, \ldots, n$ and coincides with $\tilde{g}$ otherwise, that is,

$$\tilde{g}_\perp(x) = \begin{cases} \perp, & x \in \cup_{j=t+1,\ldots,n}\mathcal{U}(x_j), \\ \tilde{g}(x), & \text{otherwise.} \end{cases}$$

Let $\mathcal{G}_n$ be the set which contains all such $\tilde{g}_\perp$. It is easy to see that $\kappa_{\mathcal{U}}(f_1), \ldots, \kappa_{\mathcal{U}}(f_k) \in \mathcal{G}_n$. Note that $\min_{g \in \mathcal{G}_n} \mathbf{Pr}\{g(x) \neq \perp\} = \Phi_{\mathcal{D}_\mathcal{X}}(n)$, we have

$$\frac{\Phi_{\mathcal{D}_\mathcal{X}}(n)}{m} \leq \mathbb{1}\{f_i(x') = y_j, \forall x' \in \mathcal{U}(x_j)\} \leq \frac{1}{m}.$$

Now for each $l$ by the linearity of expectations,

$$(t-1) \cdot \frac{k}{m} \cdot \frac{\Phi_{\mathcal{D}_\mathcal{X}}(n)}{m} \leq \mathbb{E}[M_l] \leq (t-1) \cdot \frac{k}{m} \cdot \frac{1}{m}.$$

Thus $\epsilon\,\mathbb{E}[M_l]$ holds for

$$\epsilon \leq \frac{m^2}{(t-1)k}\frac{k}{2m} = \frac{m}{2(t-1)},$$

and $\frac{\epsilon^2}{3}\mathbb{E}[M_l] \geq 3\log m$ holds for $\epsilon \geq \sqrt{9\log m \cdot \frac{m^2}{\Phi_{\mathcal{D}_\mathcal{X}}(n)(t-1)k}}$. Therefore, we can find a suitable $\epsilon$ whenever

$$\sqrt{9\log m \cdot \frac{m^2}{\Phi_{\mathcal{D}_\mathcal{X}}(n)(t-1)k}} \leq \frac{m}{2(t-1)} < 1.$$

If we choose $t = 1 + \frac{\Phi_{\mathcal{D}_\mathcal{X}}(n)k}{36\log m}$ and $k > \frac{18}{\Phi_{\mathcal{D}_\mathcal{X}}(n)}m\log m$, the aforementioned conditions hold. Note that $\mathbf{Pr}\{\kappa_{\mathcal{U}}(\hat{f}) \neq \perp\} \geq \Phi_{\mathcal{D}_\mathcal{X}}(n)$, hence the expected number of robustly and correctly predicted labels is at least

$$\frac{3}{4} \cdot t + \frac{\Phi_{\mathcal{D}_\mathcal{X}}(n)}{m}(n-t) \geq \frac{\Phi_{\mathcal{D}_\mathcal{X}}(n) \cdot n}{m} + t \cdot \left(\frac{3}{4} - \frac{\Phi_{\mathcal{D}_\mathcal{X}}(n)}{m}\right) \geq \frac{\Phi_{\mathcal{D}_\mathcal{X}}(n) \cdot n}{m} + \frac{\Phi_{\mathcal{D}_\mathcal{X}}(n) \cdot k}{144\log m},$$

which completes the proof. $\qquad\square$

## 4   Conclusion

In this work, we study the overfitting bias in the context of robust multiclass learning. We formally define the adaptive algorithms in an adversarial setting and analyze the average case performance that can be achieved by an adaptive algorithm. Upper bounds and lower bounds are both derived, and are matching within logarithmic factors when the number of test samples and distribution of data features are fixed.

## Acknowledgements

This work is supported by the National Natural Science Foundation of China under Grant 61976161, the Fundamental Research Funds for the Central Universities under Grant 2042022rc0016.

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
