# OpenReview forum: "Characterization of Overfitting in Robust Multiclass Classification"
_NeurIPS.cc/2023/Conference — NeurIPS 2023 poster_

### Official Review · Reviewer_WwD9 · 2023-07-06

**Soundness:** 3 good
**Presentation:** 4 excellent
**Contribution:** 2 fair
**Rating:** 6
**Confidence:** 4

**Summary:**

The main worry regarding the excessive reuse of test datasets in machine learning is its potential to cause overfitting. The objective of this paper is to characterize the relationship between the amount of robust overfitting bias and three key factors: the number of classes (m), the number of robust accuracy queries (k), and the size of the test dataset (n). The main theoretical contributions of this paper consist of providing upper and lower bounds on the attainable robust overfitting bias in multiclass settings.

**Strengths:**

The paper showcases exceptional writing skills, surpassing the average standard of presentation seen in many well-written and well-structured papers. The introduction successfully establishes a clear motivation and provides a satisfactory outline of the contributions. Although the section on related works is concise, it not only acknowledges a broad range of prior research efforts but also effectively highlights the key distinctions and limitations. Despite my personal unfamiliarity with several of the cited works and some of the presented topics, I never felt lost. This is mainly attributed to the "Summary of our results" section, which introduces the various notations used in the paper, provides a very clear formulation of the addressed problem, and offers a summary of the main results and the techniques employed in their proofs. The proofs of the various theoretical results are generally clear and sufficiently detailed. However, there are occasional passages that I did not understand easily and, in my opinion, require more elaboration. For instance, the usage of the Chernoff bound for Bernoulli random variables with bias could benefit from further explanation.

**Weaknesses:**

My primary concern relates to the level of novelty and originality in the theoretical results presented in the paper, particularly concerning the techniques employed in the proofs. It is worth acknowledging that the paper serves as an extension of previous works by (Feldman et al., "The advantages of multiple classes for reducing overfitting from test set reuse," ICML'19) and (Acharya & Suresh, "Optimal multiclass overfitting by sequence reconstruction from Hamming queries," ALT'20) within an adversarial setting. However, it is important to note that the proofs in this paper largely follow a similar approach to those in the aforementioned papers, and the extension to the adversarial setting does not significantly contribute to the technical aspect.

I have observed several errors in the paper (which I believe are oversights) that can be easily noticed but may lead to a misunderstanding of the contributions. One specific example is found in line 1 of Algorithm 2, where the division of n into (k-1) blocks should be denoted as (B_1...B_(k-1)) instead of (B_1, ..., b_k). It is crucial to maintain a consistent notation for the blocks using a capital B, and it should be noted that the last block is indexed as (k-1). Furthermore, on the same line, the upper and lower bounds of |B_i| need to be reversed due to the condition n > k-1. The correct bounds should be $\frac{n}{2(k-1)} \leq |B_i| \leq \frac{n}{k-1}$. These errors should be rectified to ensure the accuracy and clarity of the paper.

Another aspect that I find regrettable is the lack of discussion or interpretation regarding the various bounds presented in the paper. This absence leaves a gap in the understanding of the implications and significance of these bounds. It would have been valuable to explore the practical implications of the bounds, their relationship to existing theories or frameworks, and any potential limitations or extensions that arise from them. Providing such a discussion would have enhanced the overall comprehensiveness and depth of the paper.


**Questions:**

See the above Flaws part.

**Limitations:**

No potential negative societal impact.

---

> ### Author Rebuttal · Authors · 2023-08-06
>
> Thanks for reviewing our work and affirming our presentation; this means a lot to us. Below are our responses to your comments. We hope these address your concerns. Do not hesitate to reach out if you have further questions or suggestions.
>
> **About the novelty and originality**
>
> We feel very sorry for our inadequate referencing to known proof/techniques that may make you hard to access the originality. On one hand, the upper bounds' proof technically differs from proofs in the standard case derived by [1] from the following 3 parts:
>
> 1. To take the adversarial perturbation into account, we use Bayes' formula to represent the robust accuracy of a single sample by a Bernoulli r.v. parameterized by $C(f)$. This procedure is quite trivial in the standard case.
>
> 2. As discussed in Section 2.3, we introduce the notion of hypothesis class to upper bound $C(f)$, which makes our bounds tighter.
>
> 3. Some inequality scaling techniques, e.g., Eq.(1) in Line 125 and the inequalities at the end of the proof in Lines 132-135.
>
> On the other hand, Compared to [2],  our technical novelty can be summarized as follow:
>
> 1. To expand the domain from $\mathcal{X} _S$ to the entire $\mathcal{X},$ we borrow the idea of nearest neighbors algorithm. This construction is proved to be optimal (up to logarithmic factors) for fixed $n$ and $\mathcal{D} _\mathcal{X}$).
>
> 2. In the proof of Theorems 3 & 4. the introduction of corrupted classes is very elegant, without whom one may not be able to derive the distribution of $(N^S _{i,1},\dots,N^S _{i,m}),$ which is crucial in the standard case to prove the lower bounds. This is because the output $\hat{f}$ may **always** predict a label that is not robust to adversarial perturbation. In other words, some samples may be not to be counted. So the "always wrong" label, i.e. $\perp,$ not only facilitates our deduction, but also instruct us to study the distribution of $(N^S _{i,1},\dots,N^S _{i,m},N^S _{i,\perp}),$ which makes the counting well defined.
>
> 3. In addition, there is a step to deriving lower bounds on $\textbf{Pr}\\{\kappa _\mathcal{U}(\hat{f}(x))\neq\perp\\}$ in both proofs of Theorems 3 & 4, (Lines 174-177 & Lines 197-201) which has no parallel in previous works.
>
> 4. In our Algorithm $\mathcal{A}^{big}(C),$ the step 2.(b), we make $f _i(x _j)=\perp$. This construction makes the value of $A _l$ (Eq. (3) in Line 189) in our proof differs from that in [2], but the equation $A _{y _1}-A _l=W _0+M _{y _1}-M _l$ in Line 192 still holds so we can continue the proof. However, one can prove that this is not the case when the queries are designed such that $f _i(x _j)=1,$ as constructed in [2].
>
> **About the errors**
>
> We greatly appreciate your meticulous examination of our work and we apologize for any confusion may caused by these errors. Your postulations are totally right. We assure you that these factors do not compromise the validity of our results, and we will correct them in revision. Thanks again for your time.
>
> **About the absence of discussion**
>
> Thanks for the suggestion! We would be more than happy to add more discussion about our results and practical implications, and we would like highlight some points here. Recall that the question we focus on is: Can excessive reuse of test datasets lead to overfitting in robust learning setting? The answer to this question is implied in our Theorem 1. For fixed number of classes $m$ and fixed number of test examples $n,$ the lower and upper bounds of $h_\mathcal{U}$ is at least $\tilde{\Theta}(\sqrt{k}),$ which states that one can expect better performance with more queries made to the test set. So the answer is positive.
>
> As for the practical setting, for example, in modern ML benchmarks for adversarial robustness (e.g. RobustBench), hundreds of results have been reported on same test sets (e.g. ImageNet, SIFAR100). Large-scale hyperparameter tuning and experimental trials across numerous studies likely add thousands of queries to the test data, which may substantially "improve" robust accuracy according to our results. Our results may suggest to limit the query time or develop some monitoring mechanism to avoid malicious reuse of test sets.
>
> *Reference*
>
> *[1] Feldman Et al.  "The advantages of multiple classes for reducing overfitting from test set reuse." ICML 2019*
>
> *[2] Acharya Et al. "Optimal multiclass overfitting by sequence reconstruction from Hamming queries." ALT 2020*

---

### Official Review · Reviewer_PNTq · 2023-07-07

**Soundness:** 3 good
**Presentation:** 2 fair
**Contribution:** 2 fair
**Rating:** 5
**Confidence:** 3

**Summary:**

This paper generalizes the framework of perfect reconstruction to the adversarial setting and studies how much a k-query algorithm can overfit the test set in the adversarial setting. Upper bounds and lower bounds are derived, which match in terms of the number of classes m and the number of queries k modulo logarithmic factors when the number of test examples n and the distribution are fixed.

**Strengths:**

This paper gives a careful analysis that generalizes both the upper bound and the lower bound on how much an algorithm can overfit the test set in non-robust setting to the adversarial setting. The results are novel.

**Weaknesses:**

1. The dependence of the lower bound on the number of test examples n does not have a closed form and thus the question asked in the abstract is not fully solved.

2. It is not clear what role the perturbation set U plays in the bounds. In the adversarial setting, other than the parameters m, k and n, the radius r of the perturbation ball is also an important parameter and it is usually more informative if we know how the bounds are affected by allowing stronger attacks, i.e. a larger r. Otherwise, some intuitions on why the perturbation size is irrelevant here would be helpful.

3. This paper is in general not very well-written. For example, the term robust overfitting is confusing here: in line 9, robust overfitting [1] refers to the phenomenon that the robust test error can increase during adversarial training, but later on, e.g. in line 21 and 81, the term is used to describe the overfitting to the test set with k-query algorithms; there are not many detailed discussions on the motivation of the studied problem or what the derived bounds suggest such as how each parameter affects the bound and if robust multiclass classification requires benchmark datasets different from standard classification.


**Questions:**

See Weaknesses.

minors:
1. In line 66, should it be h_U(A) - 1/m instead?
2. In line 107, the explanation after 'i.e.' is confusing.
3. In Algorithm 1, the definition of the output is not written in a very nice way. Probably move $\forall x \in X$ outside.

**Limitations:**

Yes.

---

> ### Author Rebuttal · Authors · 2023-08-06
>
> Thank you for your time reviewing our work and your helpful suggestions.
>
> **About the dependence of lower bound on $n$**
>
> As we discuss in Lines 77-84, the term $\Phi _{\mathcal{D} _\mathcal{X}}(n)$ highly depends on the distribution of $\mathcal{D} _\mathcal{X}$ and is unavoidable. Its specific form is presented in Section 3.2. This term measures how easily to sample $n$ 'good' (for robust overfitting) test data features from $\mathcal{D} _\mathcal{X}$. For example, if the test data features are well-separated, then we have $\Phi _{\mathcal{D} _\mathcal{X}}(n)\equiv1$. But in most case the well-separated property does not hold, it can be proven that some extreme distribution of test data feature (e.g. supp($\mathcal{D} _\mathcal{X}$)$\subset\mathcal{U}(x _0)$ for some $x _0\in\mathcal{X}$) may not allow one to derive bounds w.r.t $k$ when $k\geq m$. So we keep this term in our results.
>
> **About the role $\mathcal{U}$ plays**
>
> We apologize for any confusion arising from our presentation. That is indeed an interesting issue. We first clarify that $\mathcal{U},$ or equivalently $r,$ is relevant to our results. Intuitively, bigger $r$ shrink both upper and lower bounds on $h _\mathcal{U}(k,n,m),$ and this is captured by the definitions of $C _\mathcal{H}$ and $\Phi _{\mathcal{D} _\mathcal{X}}(n)$ (see proof of Theorem 2 and definition of $\Phi _{\mathcal{D} _\mathcal{X}}(n),$ respectively). Nevertheless, we deliberately mitigate the influence of $r,$ as our focus is placed on investigating the potential for overfitting when test data is excessively reused under **specific** adversarial perturbations. Besides, our analysis suggests that the affect of $r$ is closely intertwined  with $\mathcal{D} _\mathcal{X},$ which cannot be parameterized. More explicit dependence on $\mathcal{U}$ or $r$ could potentially serve as a future avenue of exploration. Thank you again for bringing this to our attention.
>
> **About the term "robust overfitting"**
>
> Sorry for any confusion caused by our oversight. Indeed, the overfitting phenomenon in [1] can also be formulated under the $k$-query framework, since the process of adversarial training can also be viewed as adaptive queries. Nevertheless, we greatly appreciate your highlighting the potential misunderstanding to this term, and we will add some discussion in revision.
>
> **"In line 66, should it be $h_\mathcal{U}(A)-1/m$ instead?"**
>
> We sincerely appreciate your meticulous review of our paper, but there is no typo here. $h_\mathcal{U}(A;S)-1/m$ measures how much $\mathcal{A}$ robustly overfits a specific test dataset $S.$
>
> **"In line 107, the explanation after 'i.e.' is confusing."**
>
> Sorry for the confusion. The explanation means the '$\perp$' output satisfies $\perp\neq1,\dots,m$ and $\perp\neq\perp,$ see [2] for more details.
>
> **"In Algorithm 1, the definition of the output is not written in a very nice way. Probably move $\forall x\in\mathcal{X}$ outside."**
>
> Thanks again for your careful reviewing. We totally agree and will fix it in revision.
>
> *Reference*
>
> *[1] Leslie Rice Et al. "Overfitting in adversarially robust deep learning" ICML 2019*
>
> *[2] Daniel Cullina Et al. "Pac-learning in the presence of adversaries." NeurIPS 2018*

---

> > ### Comment · Reviewer_PNTq · 2023-08-17
> >
> > Thank you for the response. I believe that adding the discussions mentioned in this and other rebuttals in the final version can greatly improve the presentation of this paper. I thereby raise my score to 5.

---

> > > ### Author Response · Authors · 2023-08-18
> > > **Re: Official Comment by Reviewer PNTq**
> > >
> > > Thank you for your support! We will add more discussions to refine the presentation and enhance the quality of our submission.

---

### Official Review · Reviewer_YY54 · 2023-07-12

**Soundness:** 2 fair
**Presentation:** 1 poor
**Contribution:** 2 fair
**Rating:** 4
**Confidence:** 3

**Summary:**

In this paper, the authors consider the following question: Given the number of classes m, the number of robust accuracy queries k, and the number of test examples in the dataset with size n, how much can adaptive algorithms robustly overfit the test dataset? They solve this problem by giving upper and lower bounds of the robust overfitting accuracy in multiclass classification problems.

**Strengths:**

1. Robust classification and adversary perturbation are valuable topics in the learning theory community.
2. Some adaptive algorithms are proposed and could be useful in practice.

**Weaknesses:**

Major
1. The paper structure is not very clear. The algorithms are defined after the main results. It's very confusing.
2. Line 56-69: The definition of k-query is not reader-friendly. It's better to have some examples and figures.
3. Line 63-65: The assumption that the labels are uniformly distributed is very restricted and not general. It makes the study has limited practical and theoretical contribution.
4. The gap between the upper and lower bound is large, which makes the two bounds have limited theoretical and practical contributions.
5. In Lines 49-52, the perturbation is defined with radius r. However, in all main results, they don't contain r, and have no restriction on r. The result would be different between a tiny and a huge r.

Minor
1. The literature section needs a clearer analysis of the relationship among adaptive algorithms, overfitting, robustness, and multi-classification. In addition, the motivation is not fully explained.
2. The title "Characterization of Overfitting in Robust Multiclass Classification" is too broad. It's better to be more specific with the adaptive algorithms and adversarial settings.
3. What's mathmatical definition for \O, \Omega, and \Theta? They are not defined in the paper.
4. The main result, Theorem 1 (lines 74-76), is
labeled as informal. In addition, it's not proven and not reader-friendly.
5. The paper lacks practical experiments to support the main results.

**Questions:**

1.What's mathmatical definition for \O, \Omega, and \Theta? They are not defined in the paper.

2. Line 56-69: The definition of k-query is not reader-friendly. It's better to have some examples and figures. Could you provide a detailed explanation?

3. In Lines 49-52, the perturbation is defined with radius r. However, in all main results, they don't contain r, and have no restriction on r. The result would be different between a tiny and a huge r. Could you explain the relation between accuracy and r?

**Limitations:**

This paper doesn't include limitations and the negative societal impact of their work.

---

> ### Author Rebuttal · Authors · 2023-08-06
>
> Thank you for your time reviewing our work.
>
> **About the paper structure**
>
> We focus on the question whether adaptively excessive reuse of test data lead to overfitting in robust learning. In Section 2 , we transform this problem into a problem of studying the value of $h _\mathcal{U}(k,n,m),$ and our informal main result (Theorem 1) is a combination of our Corollary 1 and Theorems 3 & 4, which study the upper and lower bounds on $h _\mathcal{U}(k,n,m)$ and are all proven strictly.
>
> The reason why we designed the algorithms is to prove lower bounds on $h_\mathcal{U}(k,n,m).$ So we do not present the algorithms in our main results section.
>
> **About the definition of $k$-query**
>
> Sorry for the confusion. We would like to use our algorithm $\mathcal{A}^{small}$ (k>1) as an example to explain the definition of $k$-query. As we defined in Lines 56-57, $\mathcal{A}^{small}$ first makes $k$ queries, namely $f _1,\dots,f _k$ (see Algorithm 2 for specific definitions) on the test set $S.$ Note that these queries are designed in advance hence are independent to $S$. Then $\mathcal{A}^{small}$ receives the values of $\text{Acc} _\mathcal{U}(f _1;S),\dots,\text{Acc} _\mathcal{U}(f _1;S),$ and utilizes these values (robust accuracies) to construct $\mathcal{A}^{small}$'s final output $\hat{f}.$
>
> **About the assumption of uniformly distributed labels**
>
> Following [1], we make this assumptions since the algorithms have no prior knowledge about the test labels (as we state in Lines 63 & 64). Many experiments have also been conducted under this assumption, such as [3][4]. We believe this definition reflects the fact that excessive reuse of test data leads to overfitting in robust learning.
>
> **About the gap of bounds**
>
> Theorem 1 shows that our upper bounds and lower bounds are matching up to a logarithmic factor for any fixed number of test examples $n$ and feature distribution $\mathcal{D}_\mathcal{X},$ which is optimal at present.
>
> **There is no $r$ in main results**
>
> It needs to be clarified that our results definitely contain $r.$ However, to make our results more intuitive, we scale our bounds without changing its order. Specifically, in our full upper bounds, Theorem 2, $r$ affects the value of $C _\mathcal{H},$ and we scale it by the fact $C _\mathcal{H}\leq1$ in Corollary 1, that is why there is no $r$ in Theorem 1's upper bounds. On the other hand, in our formal lower bounds, namely Theorems 3 & 4, $r$ affects the value of $\Phi _{\mathcal{D} _\mathcal{X}}(n)$ explicitly (see Lines 152-154 for the definition of $\Phi _{\mathcal{D} _\mathcal{X}}(n)$, and this term is preserved in our Theorem 1's lower bounds.
>
> **The definition of $O,\Omega,\Theta$**
>
> We apologize for any confusion caused by our oversight in assuming familiarity with the definition.
>
> The $O,\Omega,\Theta$ notations, as known as the big O notations, are asymptotic notations. They are defined as follows: Given $f:\mathbb{R}\to\mathbb{R} _+$ and $g:\mathbb{R}\to\mathbb{R} _+$ we write $f=O(g)$ if there exist $x _0,\alpha\in\mathbb{R} _+$ such that for all $x>x _0$ we have $f(x)\leq\alpha g(x).$ We write $f=\Omega(g)$ if there exist $x _0,\alpha\in\mathbb{R} _+$ such that for all $x>x _0$ we have $f(x)\geq\alpha g(x).$ The notation $f=\Theta(g)$ means that $f=O(g)$ and $g=O(f).$ Finally, the notation $f=\tilde{O}(g)$ means that there exist $k\in\mathbb{N}$ such that $f(x)=O(g(x)\log^k(g(x))).$ The notation $f=\tilde{\Omega}(g)$ and $f=\tilde{\Theta}(g)$ are defined analogously.
>
> **The paper lacks practical experiments to support the main results.**
>
> Since [2] has already indicated the presence of overfitting in adversarial training. Hence our paper is devoted to a certain theoretical question related to this phenomenon, that is, whether adaptively excessive reuse of test data leads to overfitting in robust learning. Our results answer this question in the affirmative and suggest that one can expect better performance with more queries made to the test data, which aligns with the experimental results.
>
> *Reference*
>
> *[1] Vitaly Feldman Et al. "The advantages of multiple classes for reducing overfitting from test set reuse." ICML 2019*
>
> *[2] Leslie Rice Et al. "Overfitting in adversarially robust deep learning" ICML 2019*
>
> *[3] Cynthia Dwork Et al. "Generalization in Adaptive Data Analysis and Holdout Reuse" NIPS 2015*
>
> *[4] Cynthia Dwork Et al. "The reusable holdout: Preserving validity in adaptive data analysis" Science 2015*

---

> > ### Comment · Reviewer_YY54 · 2023-08-12
> >
> > Thanks very much for the replies and answers.
> >
> > 1.About the assumption of uniformly distributed labels
> >
> > As the proposed algorithm is not a Bayesian method, the assumption that the prior distribution of test labels is uniform is very strong. It makes the study has limited practical and theoretical contribution.  In the classification community, a more general distribution of test labels is a common choice.
> >
> >
> > 2.About the gap of bounds
> >
> > For example, in Theory, 1, lines 74,75, the left side contains the term \Phi_{D_X}(n). This term is only stated as <=1. In this way, it could be much smaller than 1, e.g., o(1). So, the gap could be much larger.
> >
> > 3.There is no r in the main result
> >
> > As the relation between r and C_H is not clearly stated in the theorem, so it's hard to follow the r issue.
> >
> >
> > 4.The paper lacks practical experiments to support the main results.
> >
> > For this issue, I agree with Reviewer 5K3z's comment," I understand that your work is theoretical. However, adding illustrations, at least on simulated data would bring value to the paper. It would ease the reading of the paper for a practitioner and could give ideas for future directions of research."
> >
> >
> > Therefore, I maintain my rating.

---

> > > ### Author Response · Authors · 2023-08-14
> > > **Re: Official Comment by Reviewer YY54**
> > >
> > > Thanks for your replies. In response to your initial comments, we carefully addressed each concern raised. Below, we provide a point-by-point response to your feedback:
> > >
> > > 1. About the assumption of uniformly distributed labels
> > >
> > > We'd like to kindly point out a potential oversight that we make this assumption **since** our algorithms are not Bayesian. A Bayesian method refers to an approach starting with prior distribution. It is precisely because we have no prior knowledge about the distribution of data labels, we assume they are uniformly distributed. Furthermore, we'd like to highlight again that this assumption is following [1].
> > >
> > > 2.About the gap of bounds
> > >
> > > Indeed, there might be situations as you've described, yet we would like to reiterate that this is contingent on the distribution of the data (see explicit form of $\Phi _{\mathcal{D} _{\mathcal{X}}}(n)$ is presented in Lines 152-154). For example, when $k>m$ and if the distribution of $\mathcal{D} _\mathcal{X}$ is very ill-conditioned e.g., $\text{supp}(\mathcal{D} _\mathcal{X})\subset\mathcal{U}(x _0)$ for some $x _0\in\mathcal{X},$ no algorithm would perform better than majority vote. In this case only a trivial lower bound can be derived, and we do have $\Phi _{\mathcal{D} _{\mathcal{X}}}(n)=o(1).$ That is why we consider this term as "unavoidable". Nevertheless, our lower bounds match our upper bounds for fixed $\mathcal{D} _{\mathcal{X}}$ and $n.$
> > >
> > > 3.There is no r in the main result
> > >
> > > Recall that our study focuses on studying $h _\mathcal{U}(k,n,m)$ under a given, hence fixed, adversarial perturbation, meaning our emphasis is not on $\mathcal{U}$ or $r$. The significance of $C _\mathcal{H}$ is that the upper bounds will become tighter if the algorithms are based on smaller hypothesis class.
> > >
> > > 4.The paper lacks practical experiments to support the main results.
> > >
> > > We focus on a theoretical problem related to robust overfitting, which has been indicated in numerous works. Furthermore, we would like to highlight that NeurIPS has embraced a significant number of purely theoretical contributions, e.g. [2][3[4]. In light of this, we kindly request that the reviewer consider the broader context within which our paper aligns, and we believe our contribution is in line with NeurIPS's diverse range of accepted works.
> > >
> > >
> > > *Reference*
> > >
> > > *[1] Vitaly Feldman Et al. "The advantages of multiple classes for reducing overfitting from test set reuse." ICML 2019*
> > >
> > > *[2] Ron Amit Et al. "Integral Probability Metrics PAC-Bayes Bounds" NIPS2022*
> > >
> > > *[3] Dimitris Fotakis Et al. "Linear Label Ranking with Bounded Noise" NIPS2022*
> > >
> > > *[4] Han Shao Et al. "A Theory of PAC Learnability under Transformation Invariances" NIPS2022*

---

> > > ### Author Response · Authors · 2023-08-18
> > > **Looking Forward to Your Response**
> > >
> > > We want to inquire if our rebuttal adequately addressed your concerns, as we value your feedback. We are open to further discussion if needed.
> > >
> > > best regards,

---

> > > > ### Comment · Reviewer_YY54 · 2023-08-20
> > > >
> > > > Thanks for your responses.
> > > >
> > > > Most of my concerns have been addressed. I just maintain the view about the assumption of uniformly distributed labels.
> > > >
> > > > The main result in the paper is based on the convergence rate of m. The strong assumption of uniformly distributed labels assumes that each class has a 1/m probability. I have checked that this assumption makes proof much easier, and most proof techniques only hold for uniformly distributed labels. They cannot be directly extended to a more general label distribution. This makes the theoretical contribution limited. In addition, in practice, the uniformly distributed label is also rare, which makes limited practical contributions.
> > > >
> > > > Therefore, I will raise my score to 4.

---

> > > > > ### Author Response · Authors · 2023-08-20
> > > > > **Re: Official Comment by Reviewer YY54**
> > > > >
> > > > > Thank you for investing your time and expertise in assessing our rebuttal, and we are grateful to you for the raised score. We've taken steps to provide additional clarity on this assumption. Please let us highlight again we make the uniform-label assumption since the algorithms have no prior knowledge about the test labels (as we state in Lines 63 & 64). Both theoretical and experimental results have been reported under this assumption, e.g., [1] and [2], respectively.
> > > > >
> > > > > Thanks again for your valuable review.
> > > > >
> > > > > *Reference*
> > > > >
> > > > > *[1] Vitaly Feldman Et al. "The advantages of multiple classes for reducing overfitting from test set reuse." ICML 2019*
> > > > >
> > > > > *[2] Cynthia Dwork Et al. "Generalization in Adaptive Data Analysis and Holdout Reuse" NIPS 2015*

---

### Official Review · Reviewer_5K3z · 2023-07-12

**Soundness:** 2 fair
**Presentation:** 2 fair
**Contribution:** 2 fair
**Rating:** 7
**Confidence:** 2

**Summary:**

This paper considers the problem of learning from data while being robust to the possible transformations of these data. A common practice in machine learning is to split data into a training and a test set. The latter is a holdout to evaluate the performance of the algorithm. However, recent studies have shown that reusing many times this holdout set leads to overfitting in non-robust settings. This paper questions the problem of adaptivity in an adversarial/robust setting.

**Strengths:**

The authors present a very theoretical work on robustly overfitting the test set of adaptative algorithms. They show an excellent understanding of the theory behind this problem. Remarkably, the upper bounds in the Theorem 1 match the ones of the literature when $\mathcal{U} = \mathcal{I}$ (no transformation of the set, i.e. no adversarial setting). Furthermore, for a fixed size of $S$, the test set, whose features are i.i.d. according to a fixed distribution, the upper and lower bounds match up to a logarithmic factor.

**Weaknesses:**

The paper is hard to follow for someone unfamiliar with the topic like myself. The Neurips conference gathers people from a broad audience, and this must be taken into account. In particular, no experimental part in the paper limits its impact on the community. Some numerical experiments could help understand how this work can be applied and its limits for future directions. Adding a toy numerical example could also motivate us why we should care about the question the authors ask in the abstract.

**Questions:**

Suggestions:
- recall what are $O, \Omega$ and $\Theta$
- define $\widetilde{O}, \widetilde{\Omega}, \widetilde{\Theta}$ even if it seems to be trivial for the authors,
- add an experimental section with experiments at least on simulated data and, better, on real datasets to illustrate the different theorems.

Question:
- are these $\widetilde{O}, \widetilde{\Omega}, \widetilde{\theta}$ a classical tool in your community? Why not expose the full bound with the logarithm factor? Does it help the interpretation?

**Limitations:**

No limitations are declared in the paper. The paper is theoretical and seems far from having a possible negative impact.

---

> ### Author Rebuttal · Authors · 2023-08-06
>
> Thank you for your for reviewing our paper. It is worth acknowledging that numerous studies have already indicated the presence of overfitting phenomena in adversarial training, e.g.,[1]. To this end, our paper is devoted to a certain **theoretical** question related to this phenomenon, that is, whether adaptively excessive reuse of test data leads to overfitting in robust learning. Our results answer this question in the affirmative and suggest that one can expect better performance with more queries made to the test data.
>
> The $O,\Omega,\Theta$ notations, as known as the big O notations, are asymptotic notations. we occasionally use them to clarify the main results. They are common in computer science & machine learning community. They are defined as follows:
>
> Given $f: \mathbb{R} \to \mathbb{R} _{+}$ and $g: \mathbb{R} \to \mathbb{R} _{+}$ we write $f=O(g)$ if there exist $x _0,\alpha\in\mathbb{R} _+$ such that for all $x>x _0$ we have $f(x)\leq\alpha g(x).$ We write $f=\Omega(g)$ if there exist $x _0,\alpha\in\mathbb{R} _+$ such that for all $x>x _0$ we have $f(x)\geq\alpha g(x).$ The notation $f=\Theta(g)$ means that $f=O(g)$ and $g=O(f).$ Finally, the notation $f=\tilde{O}(g)$ means that there exist $k\in\mathbb{N}$ such that $f(x)=O(g(x)\log^k(g(x))).$ The notation $f=\tilde{\Omega}(g)$ and $f=\tilde{\Theta}(g)$ are defined analogously.
>
> *Reference*
>
> *[1] Leslie Rice Et al. "Overfitting in adversarially robust deep learning" ICML 2019*

---

> > ### Comment · Reviewer_5K3z · 2023-08-12
> >
> > I thank you to have answered my questions.
> >
> > I understand that your work is theoretical. However, adding illustrations, at least on simulated data would bring value to the paper. It would ease the reading of the paper for a practitioner and could give ideas for future directions of research.
> >
> > Therefore, I maintain my rating.

---

> > > ### Author Response · Authors · 2023-08-14
> > >
> > > Thank you for your reply. However, it seems there might have been a misunderstanding in my previous response. We focus on a theoretical problem related to robust overfitting, which has been indicated in numerous works. Furthermore, we would like to highlight that NeurIPS has embraced a significant number of purely theoretical contributions, e.g. [1][2][3]. In light of this, we kindly request that the reviewer consider the broader context within which our paper aligns, and we believe our contribution is in line with NeurIPS's diverse range of accepted works.
> > >
> > > Thank you again for your time.
> > >
> > > *Reference*
> > >
> > > *[1] Ron Amit Et al. "Integral Probability Metrics PAC-Bayes Bounds" NIPS2022*
> > >
> > > *[2] Dimitris Fotakis Et al. "Linear Label Ranking with Bounded Noise" NIPS2022*
> > >
> > > *[3] Han Shao Et al. "A Theory of PAC Learnability under Transformation Invariances" NIPS2022*

---

> > > > ### Comment · Reviewer_5K3z · 2023-08-14
> > > >
> > > > Good references. Since it was my major concern, I decided to update my rating to a 7.

---

### Official Review · Reviewer_JGWq · 2023-07-17

**Soundness:** 3 good
**Presentation:** 3 good
**Contribution:** 2 fair
**Rating:** 5
**Confidence:** 2

**Summary:**

This paper studies bounds on the maximal difference between the robust accuracy of a classifier on the test set. Here, robustness means when each instance is allowed to move within a given radius within an $L_p$ ball, and the maximal difference is obtained with respect to $1/m$, where $m$ is the number of classes.

The classifier's accuracy is obtained through querying an algorithm, and lower/upper bounds are obtained in terms of the number of queries made, the number of elements in the test set and the number of classes.

**Strengths:**

* A concise paper that, for once, does not have tons of supplementary material.

* A rather rigorous treatment of the problem, with the paper consisting almost entirely of providing the proofs of the results. I think the results presented are correct (but this is not in my main area of expertise, so I could have missed something, and it would be necessary to be checked by a more expert reviewer foxued on this area of research)

**Weaknesses:**

* Better motivation of the results through illustration/practical problem mention: the paper focuses on providing a rigorous bound proof for a particular setting (robust overfitting), and assumes that the reader is convinced of the importance of the problem. In this sense it is quite focused, and it would have been nice for the reader to have at least some indication of the practical setting where this result could be used, or even better a little illustrative (synthetic) example of the situation one would like to deal with. This could be done, e.g., by dropping Lemma 1 proof if this one is available elsewhere (Lemma 4 in [6], according to the paper), as it takes almost one pages.

* Clarifications about the assumptions and their limitations: in order to obtain the provided results, the paper makes some key assumptions whose limitations and need should be discussed in more details. Two main examples of this are the following:

   * Well-separatedness of the classes is mentioned P3, L3 (note: it is not mentioned this way in reference [14]) seems equivalent to assume a dceterministic relationship between input and classes, with furthermore the requirement to be separated by "large margins", so basically assuming that one is working under a version space assumption. There would be a need to clarify that, to discuss how realistic this is, and how it can impact the provided results

  * Uniformity of the labels: another assumptions is that the output labels are perfectly uniformly distributed. This is unrealistic in most practical settings, and it is not clear how the results would change in case of imbalanced classes (goign from a slight departure of this prior information to a large one).



**Questions:**

Questions
---

(wrap up weakness points for some)

* Would it be possible to provide a slight illustration of the discussed results, or to give pointers to practical uses of these results?

* The idealistic assumptions made in the paper (well-separatedness, label uniformity) are rarely met in practice. It is not clear how limiting this is, would one apply the obtained bounds in a practical setting?

* I did not find in the paper a discussion about the quality of the provided bounds: can they expected to be tight? How useful are they likely to be in practice?

Suggestions
---

* Some typos remain in the paper (e.g., in Theorem 2: an perturbation, $b=k ln(n+1)1$, L171: "we will show it later than")

* Some parts of the proofs and technical aspects could be clarified by adding some steps that I missed while reading it. As examples:
   * P2, L39: it may be obvious, but what is the "closure" here (are we speaking about a specific closure operator?)
   * P5, L131: the derivation of $m\epsilon \geq 2 \geq 2 \mathcal{C}_\mathcal{H}$ is not immediate to me, maybe add a step?
   * P6, L161-L162: I do not see how we can obtain the numbers of correctly predicted labels only depend on the $N_m$, and also why it should only depends on $N_1,N_2$?


**Limitations:**

See weaknesses and questions comments about the initial assumptions.

---

> ### Author Rebuttal · Authors · 2023-08-06
>
> Thank you for reviewing our work and your considerate suggestions. Here are responses to your concerns.
>
> **About the results.**
>
> We apologize for our presentation. Motivated by [1] and [2], the question we focus on is: Can excessive reuse of test datasets lead to overfitting in robust learning setting? The answer to this question is implied in our Theorem 1. For fixed number of classes $m$ and fixed number of test examples $n,$ the lower and upper bounds of $h_\mathcal{U}$ is at least $\tilde{\Theta}(\sqrt{k}),$ which states that one can expect better performance with more queries made to the test set. So the answer is positive.
>
> As for the practical setting, for example, in modern ML benchmarks for adversarial robustness (e.g. RobustBench), hundreds of results have been reported on same test sets (e.g. ImageNet, SIFAR100). Large-scale hyperparameter tuning and experimental trials across numerous studies likely add thousands of queries to the test data, which may substantially "improve" robust accuracy according to our results. Our results may suggest to limit the query time or develop some monitoring mechanism to avoid malicious reuse of test sets, which, however, is far beyond the scope of this article.
>
> All in all, thank you for the suggestion, we will add some discussion about this part in revision.
>
> **About the assumptions**
>
> Sorry again for the confusion caused by our presentation.
>
> 1. Actually we do not make the well-separated assumption. We mention this concept to emphasize that the term $\Phi_{\mathcal{D}_\mathcal{X}}(n)$ is unavoidable if we make no assumption on the distribution of the data feature. In fact, one can easily show that under the well-separated assumption, our setting degenerates into standard perfect label reconstruction problem.
> 2. Following [2], we assume the label to be uniform distributed since the algorithms have no prior knowledge about the test labels (as we state in Lines 63 & 64). Many experiments have also been conducted under this assumption, such as [3][4].
>
> **About the quality of our bounds**
>
> The quality of our bounds are discussed in Lines 83-84: "Note that for a fixed  size of  $S$ whose features are i.i.d. according to a fixed distribution $\mathcal{D}_\mathcal{X}$, the upper and lower bounds  are matching up to a logarithmic factor."
>
> **About the typos and suggestions on proofs and technical aspects**
>
> We appreciate your careful review of the article and point these typos out, we will fix them in our final revision. The following is our response to your suggestions one by one:
>
> 1. Thank you for your suggestion, we will define the "support" more explicitly in revision.
> 2. The derivation $m\epsilon\geq2$ is obtained by setting $\epsilon=\frac{2b}{n}$ (see the second formula in Theorem 2), and we do miss it in our proof. We will perfect this proof in revision.
> 3.  The numbers of correctly predicted labels only depend on $N_1$ and $N_2$ follows from the construction of our Algorithm 1, whose output $\hat{f}$ only predicts labels $1$ and $2.$
>
> *Reference*
>
> *[1] Vitaly Feldman Et al. "Open problem: How fast can a multiclass test set be overfit?" Colt 2019*
>
> *[2] Vitaly Feldman Et al. "The advantages of multiple classes for reducing overfitting from test set reuse." ICML 2019*
>
> *[3] Cynthia Dwork Et al. "Generalization in Adaptive Data Analysis and Holdout Reuse" NIPS 2015*
>
> *[4] Cynthia Dwork Et al. "The reusable holdout: Preserving validity in adaptive data analysis" Science 2015*

---

> > ### Comment · Reviewer_JGWq · 2023-08-19
> > **Thanks for the answer, still unclear about the practical setting.**
> >
> > Dear authors,
> >
> > Thank you very much for your answers and clarification. I have also read the other reviewer comments, and a number of them share the same concerns about the practical impact of the presented results: while I am not at all against fully theoretical results, I do think that outside of pure mathematics (and Neurips is not about pure mathematics) one should at least have an idea of how the theory should serve the practice, or why filling a theoretical hole may be helpful.
> >
> > In light of this, I am still yet not fully convinced of the arguments brought forward, which mainly conist (if I am correct) that it is less robust to always test on the same test set (for it can lead to optimize for this test sets). Overall this makes sense, but in my opinion this is largely a problem created by academic ML and the way algorithms are benchmarked, and in this sense it is hard to consider theoretical results about this being of practical importance in actual ML applications. Most ML practitioners would not rely on a single test set to assess the value of a method.
> >
> > Right now I will keep my score, which is positive as I appreciated the paper that looks at waht appears to be a hard problem, but I must say that for the moment I am unable to appreciate its significance.
> >
> > Best regards

---

> > > ### Author Response · Authors · 2023-08-19
> > > **About the practical setting**
> > >
> > > Dear Reviewer JGWq,
> > >
> > > Thank your for your response! We kindly emphasize again that, as highlighted in **About the results** section in our rebuttal, our results do have practical scenery, e.g., RobustBench[1] (A authoritative benchmarking platform designed to evaluate the robustness of machine learning models), where the test sets are fixed and thousands of results have been reported on them. As we discussed, our theory could potentially offer insights for the design of the monitoring mechanism of platforms, or at the very least, serve as a reminder. It is also noteworthy that much like the those in RobustBench, benchmarks in modern machine learning are in an adaptive fashion, where new models can be dependent on the previous models. This is the reason we choose our setting, and we think our contribution is in line with NeurIPS's diverse range of accepted works.
> > >
> > > Best,
> > >
> > > *Reference*
> > >
> > > *[1] https://robustbench.github.io/*

---

### Official Review · Reviewer_UhYx · 2023-07-25

**Soundness:** 4 excellent
**Presentation:** 3 good
**Contribution:** 3 good
**Rating:** 6
**Confidence:** 4

**Summary:**

The authors present near-matching lower and upper bounds for the robust accuracy of an adaptive algorithm having a budget of k accesses to the oracle of accuracy on a test set of size n and m classes.

**Strengths:**

The proof is really clear and pedagogical. The related work is clear.

**Weaknesses:**

The problem is not well sold nor introduced. I had to look at the problem formulation to understand what the authors were intending to do. While related work is abundant, more space can be used for explaining what are the goals of the paper for new readers. And the conclusion could be more than just a summary, starting future work.

Also, as the authors are humble, they are borrowing tools from several existing proofs so it is not clear whether this is a breakthrough or incremental work. Still, it is nicely put.

Minor comments:
"binominal" => binomial

**Questions:**

No

---

> ### Author Rebuttal · Authors · 2023-08-06
>
> We sincerely appreciate the time you've dedicated to reviewing our work. We would like to highlight the originality of our proof techniques, which are outlined as follows.
>
> On one hand, the upper bounds' proof technically differs from proofs in the standard case derived by [1] from the following 3 parts:
>
> 1. To take the adversarial perturbation into account, we use Bayes' formula to represent the robust accuracy of a single sample by a Bernoulli r.v. parameterized by $C(f)$. This procedure is quite trivial in the standard case.
>
> 2. As discussed in Section 2.3, we introduce the notion of hypothesis class to upper bound $C(f)$, which makes our bounds tighter.
>
> 3. Some inequality scaling techniques, e.g., Eq.(1) in Line 125 and the inequalities at the end of the proof in Lines 132-135.
>
> On the other hand, Compared to [2],  our technical novelty can be summarized as follow:
>
> 1. To expand the domain from $\mathcal{X} _S$ to the entire $\mathcal{X},$ we borrow the idea of nearest neighbors algorithm. This construction is proved to be optimal (up to logarithmic factors) for fixed $n$ and $\mathcal{D} _\mathcal{X}$).
>
> 2. In the proof of Theorems 3 & 4. the introduction of corrupted classes is very elegant, without whom one may not be able to derive the distribution of $(N^S _{i,1},\dots,N^S _{i,m}),$ which is crucial in the standard case to prove the lower bounds. This is because the output $\hat{f}$ may **always** predict a label that is not robust to adversarial perturbation. In other words, some samples may be not to be counted. So the "always wrong" label, i.e. $\perp,$ not only facilitates our deduction, but also instruct us to study the distribution of $(N^S _{i,1},\dots,N^S _{i,m},N^S _{i,\perp}),$ which makes the counting well defined.
>
> 3. In addition, there is a step to deriving lower bounds on $\textbf{Pr}\\{\kappa _\mathcal{U}(\hat{f}(x))\neq\perp\\}$ in both proofs of Theorems 3 & 4, (Lines 174-177 & Lines 197-201) which has no parallel in previous works.
>
> 4. In our Algorithm $\mathcal{A}^{big}(C),$ the step 2.(b), we make $f _i(x _j)=\perp$. This construction makes the value of $A _l$ (Eq. (3) in Line 189) in our proof differs from that in [2], but the equation $A _{y _1}-A _l=W _0+M _{y _1}-M _l$ in Line 192 still holds so we can continue the proof. However, one can prove that this is not the case when the queries are designed such that $f _i(x _j)=1,$ as constructed in [2].
>
> We will add more discusstion about this part in revision. Thank you again for checking this out.
>
> *Reference*
>
> *[1] Vitaly Feldman Et al. "The advantages of multiple classes for reducing overfitting from test set reuse." ICML 2019*
>
> *[2] Acharya Et al. "Optimal multiclass overfitting by sequence reconstruction from Hamming queries." ALT 2020*

---

> > ### Comment · Reviewer_UhYx · 2023-08-16
> >
> > Thanks. I also enjoyed the discussion between the authors and reviewer WwD9. I think that this work should be more sold, more contextualized, get more perspective to reach a broader audience.
> >
> > (Hyperparameter tuning is supposed to be made on a validation set, not the test set. But indeed when a test set is public you never know what made practitioners choose a hyper-parameter compared to another. You could for example mention that test set could leak into the hyper-parameters.)

---

> > > ### Author Response · Authors · 2023-08-17
> > > **Re: Official Comment by Reviewer UhYx**
> > >
> > > Thanks for the support and your considerate suggestions! We will add more discussion to perfect our work.

---

### Decision · Program_Chairs · 2023-09-21

**Decision:**

Accept (poster)

**Comment:**

This meta review is based on the reviews, the authors rebuttal and the discussions with the reviewers, discussions with the SAC, and ultimately my own judgement on the paper. There was a majority to feel that the paper contributes sound and interesting contributions. I feel this work deserves to be featured at NeurIPS and will attract interest from the community, although I would like to personally invite the authors to carefully revise their manuscript to take into account the remarks and suggestions made by reviewers. Congratulations!